# Comparative Physiological and Transcriptome Analysis Reveals Potential Pathways and Specific Genes Involved in Waterlogging Tolerance in Apple Rootstocks

**DOI:** 10.3390/ijms24119298

**Published:** 2023-05-26

**Authors:** Kunxi Zhang, Xiaofei Chen, Penghao Yuan, Chunhui Song, Shangwei Song, Jian Jiao, Miaomiao Wang, Pengbo Hao, Xianbo Zheng, Tuanhui Bai

**Affiliations:** College of Horticulture, Henan Agricultural University, Zhengzhou 450002, China; kunxi66@163.com (K.Z.); chenxiaofei0312@163.com (X.C.); q18737360911@163.com (P.Y.); songchunhui060305@126.com (C.S.); jiaojian@henau.edu.cn (J.J.); wmm2018@henau.edu.cn (M.W.); hao_pb@163.com (P.H.)

**Keywords:** waterlogging stress, apple rootstocks, flavonoids, RNA-seq, ethylene

## Abstract

Apple (*Malus × domestica* Borkh.) is one of the most cultivated fruit crops in China. Apple trees frequently encounter waterlogging stress, mainly due to excess rainfall, soil compaction, or poor soil drainage, results in yellowing leaves and declined fruit quality and yield in some regions. However, the mechanism underlying the response to waterlogging has not been well elucidated. Therefore, we performed a physiological and transcriptomic analysis to examine the differential responses of two apple rootstocks (waterlogging-tolerant *M. hupehensis* and waterlogging-sensitive *M. toringoides*) to waterlogging stress. The results showed that *M. toringoides* displayed more severe leaf chlorosis during the waterlogging treatment than *M. hupehensis.* Compared with *M. hupehensis*, the more severe leaf chlorosis induced by waterlogging stress in *M. toringoides* was highly correlated with increased electrolyte leakage and superoxide radicals, hydrogen peroxide accumulation, and increased stomata closure. Interestingly, *M. toringoides* also conveyed a higher ethylene production under waterlogging stress. Furthermore, RNA-seq revealed that a total of 13,913 common differentially expressed genes (DEGs) were differentially regulated between *M. hupehensis* and *M. toringoides* under waterlogging stress, especially those DEGs involved in the biosynthesis of flavonoids and hormone signaling. This suggests a possible link of flavonoids and hormone signaling to waterlogging tolerance. Taken together, our data provide the targeted genes for further investigation of the functions, as well as for future molecular breeding of waterlogging-tolerant apple rootstocks.

## 1. Introduction

Global climate change has increased the occurring frequency of extreme weather events, such as heavy rainfall. Together with poor soil drainage and irrigation, frequent and heavy rainfall in local areas often leads to flooding events [1]. As a consequence, flooding is more frequent and unpredictable and has already become one of the major abiotic stresses for plants [1,2]. In fact, flooding encompasses two forms: one is referred to as “waterlogging stress”, where only the root tissue is covered by water, and the other is named “submergence”, where partial or whole shoots are under water [3]. Given the fact that the diffusion rate of oxygen in water usually drops 104 times more than in air, the excess water of waterlogging or submergence disrupts the oxygen supply from air to plant [4,5]. The lower oxygen availability renders plants hypoxic or anoxic, which ultimately disrupts several physiological activities of plants, leading to abnormal growth and development, eventually resulting in severe yield loss and large economic loss [2,5,6,7]. Therefore, a comprehensive study of the mechanism of plant response to waterlogging or submergence stress tolerance is necessary.

Intensive work have been performed in studying the effects of waterlogging on plants [3,7,8,9,10,11,12,13,14]. Many physiological activities of plants are affected because of the lack of oxygen supply under excess water conditions. One obvious characteristic is the partial or complete inhibition of root aerobic respiration, for which plant cells switch to process fermentation [15,16,17]. Fermentation results in a strong reduction in ATP synthesis and energy supply, which has severe effects on root development, nutrient uptake, and a wide range of intracellular biochemical reactions and physiological processes [18]. During long-term waterlogging, fermentation also results in the accumulation of toxic compounds such as ethanol and acetaldehyde, strongly impairing plant growth and development [19,20]. Furthermore, insufficient oxygen supply also limits photosynthesis. This is linked to the stomata closure, reduced CO_2_ absorption, leaf senescence, and chlorosis under waterlogging stress, leading to a reduced photosynthesis rate [14,21,22,23]. Moreover, reactive oxygen species (ROS) are also accumulated under waterlogging stress, leading to the lipid peroxidation of the leaf cell membrane and oxidative damage of proteins and DNA [5,24,25,26,27,28].

To adapt, plants have evolved different mechanisms. One obvious morphological adaptation is the formation of aerenchyma under waterlogged conditions. This phenomenon has been observed in many species, including rice [29], maize [30], soybean [31], evergreen trees *Luma apiculate* and *Drimys winteri* [32], and fruit trees, such as *Annona* [33] and *Prunus* ‘*Mariana 2624*’ [34]. Aerenchyma formation helps plants escape from the anoxic condition and increases the contact of plants to air, thus providing an alternative pathway to processing gas exchange and transporting oxygen to anoxic tissues under waterlogged conditions, ensuring a normal physiological metabolism and growth and development [5,7,35]. In addition, the ability to mitigate the damage of ROS overproduction, or to alleviate the limitation of photosynthesis, accounts for other important adaptation mechanisms of plants to waterlogging conditions. For example, compared with waterlogging-sensitive cherry rootstock *P. mahaleb*, the ROS-scavenging enzymes such as catalase (CAT), peroxidase (POD), and glutathione reductase (GR) are highly activated in the tolerant cherry rootstocks *P. pseudocerasus* and *P. cerasus × P. canescens* to keep the balance of ROS and thus sustain normal root activity under short-term waterlogging stress [28]. The tolerant sorghum cultivar Jinuoliang01 displayed a higher net photosynthetic rate, which is attributed to the higher chlorophyll content, greater peroxidase and catalase activities, more stable PSII, and intact chloroplast structure under waterlogging stress [14]. Exogenous application of melatonin improves the resistance of apple rootstock seedlings to waterlogging stress, possibly by the induction of aerobic respiration, preservation of photosynthesis, and alleviation of oxidative damage [23].

Apart from the abovementioned adaptation mechanisms, phytohormones, such as abscisic acid (ABA), gibberellins (GAs), brassinosteroid (BRs), ethylene, and auxin, are also the key elements in regulating the waterlogging tolerance of plants. Among those endogenous phytohormones, ethylene has been identified as the primary signal in regulating plant waterlogging tolerance [5,11,36,37]. The biosynthesis from precursor 1-Aminocyclopropane-1-carboxylic acid (ACC) to ethylene is catalyzed by ACC oxidase (ACO), which requires the participation of O_2_. Under waterlogged conditions that deplete O_2_, this process is stopped and ACC moves from the waterlogged root upward to the aerobic part of plants to synthesize ethylene [38]. Since ethylene is a gaseous hormone and its diffusion is dropped due to excess water, ethylene is more accumulated [39]. The rapid accumulation of ethylene activates the ethylene signaling pathway and induces the crosstalk between other phytohormone signaling pathways, finally leading to physiological and morphological adaptations to waterlogging stress [5,33]. In maize, ethylene can induce the expression of ZmEREB180, a member of group VII ethylene-response factors (ERFVIIs), and the overexpression of ZmEREB180 improves the plant’s long-term waterlogging stress tolerance by regulating the formation of adventitious roots and ROS homeostasis [13]. OsEIL1a, an ET-responsive transcription factor, is regulated by ethylene and promotes the synthesis of GA_4_ by directly binding to the promotor of the GA biosynthesis gene SEMIDWARF1 (SD1), and thus it may positively regulate the elongation of rice internode under waterlogging stress [11]. Most works have been conducted explicitly to understand the mechanism of waterlogging stress on the model plants Arabidopsis and rice; the mechanism may differ in different species.

Apple (*Malus domestica* Borkh.) is one of the most economically important fruit trees in the world. However, apple orchards often suffer from waterlogging stress due to heavy rainfall in the summer and poor soil drainage and irrigation practice, which strongly affect apple quality and yield. As a perennial woody plant, the resistance of apple trees to abiotic stress is largely dependent on the ability of rootstocks [40]. *M. hupehensis* and *M. toringoides*, which originated in China, have high genetic stability because of apomixes and are often used as experimental materials for comparative analysis of apple rootstock stress tolerance [41,42]. Our previous study showed that *M. hupehensis* was tolerant to waterlogging, while *M. toringoides* was more sensitive to waterlogging [41]. Using these two apple rootstocks, we performed a comparative study to explore the potential differences in their physiological responses and transcriptome profiles to waterlogging stress, aiming to identify novel genes involved in the waterlogging tolerance of apple and also reveal its possible underlying mechanism.

## 2. Results

### 2.1. Differences in Growth and Morphology between M. toringoides and M. hupehensis under Waterlogging Stress

As a first step, we probed for a potential difference in waterlogging tolerance between *M. toringoides* and *M. hupehensis* following 6 days of waterlogging stress (Figure 1). In the absence of waterlogging stress, both *M. toringoides* and *M. hupehensis* showed normal growth with upright stems and green leaves. Notably, no obvious phenotypical changes were observed in *M. hupehensis* during all 6 days of waterlogging stress. In comparison, after 3 days of waterlogging stress, most *M. toringoides* started to display a clear leaf chlorosis phenotype, characterized by wilted and drooping young leaves, while the mature leaves remained unaffected. Six days of waterlogging stress further enhanced the damage in *M. toringoides*, where the stem became slightly bent and severely wilted, drooping phenotypes were observed in both young and mature leaves, and the young leaves even became dead and yellow. Thus, *M. hupehensis* displayed more tolerance than *M. toringoides* in response to the waterlogging treatment.

### 2.2. Differences in Electrolyte Leakage and Relative Water Content Changes between M. toringoides and M. hupehensis under Waterlogging Stress

Waterlogging treatment could induce an increase in electrolyte leakage (EL) in both *M. toringoides* and *M. hupehensis* (Figure 2A,B). However, we observed significant differences between *M. toringoides* and *M. hupehensis* with respect to the amplitude: the EL of untreated *M. hupehensis* (CK) was maintained at around 50~55% in the following 5 days. Upon waterlogging treatment, the EL of *M. hupehensis* (waterlogging) increased quickly on the first day of waterlogging stress by about 8% and returned to the initial value on the second day, but kept increasing in the following three days and finally reached the maximum on the fifth day (around 65%) (Figure 2A). In comparison, the waterlogging-induced increase in EL developed much faster in *M. toringoides*, although the EL of the untreated *M. toringoides* was maintained at around 60% in the following 5 days, that showed a slightly higher value than untreated *M. hupehensis* (CK) (Figure 2B). For waterlogging-treated *M. toringoides*, the EL (waterlogging) increased sharply to around 80% after waterlogging treatment and was maintained around 80% in the following days, although there was a slight decrease on the second day of the waterlogging treatment (Figure 2B). Thus, *M. hupehensis* displayed a slow increment of waterlogging-treatment-induced electrolyte leakage.

A similar pattern was also observed in the leaf relative water content between *M. toringoides* and *M. hupehensis* after waterlogging treatment (Figure 2C,D). For *M. hupehensis*, while the leaf relative water content of the untreated group was maintained at around 80% during the period, the leaf relative water content of waterlogging-treated *M. hupehensis* remained unaffected on the first two days at around 80%, decreased to the lowest level of 40% on the third day, then returned to 50% and stayed constant on the following two days (Figure 2C). In comparison, the waterlogging treatment severely affected the leaf relative water content of *M. toringoides* (Figure 2D); its relative water content remained unchanged on the first day of the waterlogging treatment, but continuously and sharply decreased from 80% on the second day to 20% of the fifth day of the waterlogging treatment, compared with the untreated *M. toringoides*, which stayed around 60~80%. Therefore, the leaf relative water content of *M. hupehensis* was less affected by the waterlogging treatment compared with that of *M. toringoides*.

### 2.3. Differences in O_2_^•−^ and H_2_O_2_ Accumulations between M. toringoides and M. hupehensis under Waterlogging Stress

Since waterlogging can induce the reactive and toxic ROS accumulations that lead to oxidative stress and ultimately cause leaf death [27,28], we wondered if the differences in reactive and toxic ROS accumulations would contribute to the observed morphological differences in leaves between *M. toringoides* and *M. hupehensis*. To test it, NBT and DAB staining were adopted to dissect O_2_^•−^ and H_2_O_2_ accumulations, respectively. The number or area of blue or reddish-brown precipitates represents the levels of toxic O_2_^•−^ and H_2_O_2_ accumulations. By this strategy, we observed that no obvious blue dots were seen in the untreated leaves of *M. toringoides* and *M. hupehensis* (Figure 3). However, a dense and large number of blue dots fully filled in the whole leaves of waterlogging-treated *M. toringoides*, while fewer and more sparse blue dots were distributed in the leaves of waterlogging-treated *M. hupehensis* (Figure 3). This was similar to the DAB staining results (Figure 3). Larger, reddish-brown areas were clearly seen in the leaves of waterlogging treated *M. toringoides* compared with those of *M. toringoides*, where only four clear reddish-brown dots were visible (Figure 3). Taken together, these results indicate that O_2_^•−^ and H_2_O_2_ were more accumulated in the leaves of waterlogging-treated *M. toringoides* than those of *M. hupehensis*.

### 2.4. Differences in Stomatal Behavior and the Leaf Maximal Photochemical Efficiency (Fv/Fm) between M. toringoides and M. hupehensis under Waterlogging Stress

Stomata closure has been well documented as an important responsive reaction of plants to waterlogging stress [43]. This was indeed observed in the present study, where waterlogging treatment could induce the stomata closure of both *M. toringoides* and *M. hupehensis* (Figure 4A). However, the extent of stomata closure was different between *M. toringoides* and *M. hupehensis*, as observed by a clear difference in the stomata size and density. For *M. toringoides*, the length, width, and density were significantly decreased by waterlogging stress from 19.03 μm to 15.97 μm, 3.22 μm to 1.15 μm, and 3.8/mm^2^ to 1.9/mm^2^, respectively (Figure 4B–D). In contrast, the reduction was less pronounced in *M. hupehensis*, where the length and width decreased only from 11.96 μm to 10.90 μm and from 1.57 μm to 1.30 μm, respectively, while the density of stomata showed no significant changes under waterlogging stress (Figure 4B–D). These results suggest that *M. toringoides* is sensitive to waterlogging stress.

Waterlogging stress also decreased the leaf maximal photochemical efficiency (*Fv*/*Fm*) in *M. toringoides* and *M. hupehensis* (Figure 5): waterlogging stress significantly reduced the *Fv*/*Fm* of *M. toringoides* to 0.7, compared to 0.8 in the control group. In comparison, the *Fv*/*Fm* of *M. hupehensis* was only slightly affected and reduced by the waterlogging stress.

### 2.5. Differences in Ethylene Production between M. toringoides and M. hupehensis under Waterlogging Stress

The ethylene production in untreated *M. toringoides* and *M. hupehensis* almost stayed constant and was at the same level (Figure 6). After waterlogging stress, the ethylene production in *M. toringoides* and *M. hupehensis* was highly induced, while the overall level of ethylene production in the waterlogged *M. toringoides* was much higher than that of the waterlogged *M. hupehensis*. Interestingly, waterlogging-stress-induced ethylene production behaved differently between *M. toringoides* and *M. hupehensis* (Figure 6). In *M. toringoides*, the waterlogging-stress-induced ethylene production peaked twice: at 9 h and 48 h to 96 h of waterlogging stress, respectively, at which time points the level of ethylene production was around 5 times and 2~3 times higher that of *M. hupehensis*, respectively. In comparison, the ethylene production of *M. hupehensis* was gradually induced and increased in the first 48 h, then declined during the subsequent period of waterlogging treatment. Based on these results, we speculated that under waterlogging stress, ethylene acted not only as a signal but also as a toxic hormone in *M. toringoides*, while in *M. hupehensis*, ethylene only acted as a signal compound.

### 2.6. Transcriptome Sequencing and Mapping to the Reference Genome

To explore the potential differences in the mechanism of waterlogging-tolerant *M. hupehensis* and the waterlogging-sensitive *M. toringoides* in response to waterlogging stress, twelve libraries (each containing three replicates) from the leaf tissue of the control group *M. toringoides* (TCK) and *M. hupehensis* (HCK) and the waterlogged group *M. toringoides* (TW) and *M. hupehensis* (HW) were constructed and sequenced by Illumina HiSeq. After removing the low-quality reads, approximately more than 22 million clean reads with more than 6.8 billion clean bases, 150 in length, were obtained for each library. For each library, the Q20 and Q30 levels of the clean reads were above 97.8% and 94.4%, respectively, and the GC ratios were more than 47.8% (Appendix A). After mapping them to the apple reference genome using TopHat 2, we found that the mapped ratios of each sequenced library were in the range of 73.50–78.45%. Additionally, the uniquely mapped ratios were in the range of 59.37–65.81%, while the multiple mapped reads were among 12.02–15.37% (Appendix A).

### 2.7. Identifying the Differentially Expressed Genes (DEGs)

To identify the differences in gene expression between *M. toringoides* and *M. hupehensis* in response to waterlogging stress, we compared the DEGs in the four groups (HCK-vs.-HW, TCK-vs.-TW, HCK-vs.-TCK, and HW-vs.-TW) (Figure 7). For each group, the number of upregulated genes was 5110, 3204, 4794, and 7419, respectively, and the number of downregulated genes was 3732, 4321, 3863, and 6494 (Figure 7A). Additionally, we observed more DEGs in *M. hupehensis* during waterlogging stress compared with those in *M. toringoides*, suggesting that *M. hupehensis* displayed a stronger transcriptional response to waterlogging stress than *M. toringoides* (Figure 7A). A Venn diagram was further constructed to analyze the abovementioned DEGs (Figure 7B). The DEGs can be classified into two main types: genotype-specific and common waterlogging-stress-responsive genes. We observed a total of 1602 (HCK-vs.-HW) and 1463 (TCK-vs.-TW) genotype-specific responsive genes and 2614 common waterlogging-stress-responsive genes (HCK-vs.-HW and TCK-vs.-TW). Considering the genetic background difference between *M. hupehensis* and *M. toringoides*, the subsequent transcriptome analysis was mainly focused on the comparisons of TCK-vs.-TW and HCK-vs.-HW.

### 2.8. GO and KEGG Pathway Enrichment Analysis

To fully understand the role of the abovementioned DEGs between *M. toringoides* and *M. hupehensis* in response to waterlogging stress, we performed a GO functional enrichment analysis. As a result, in both TCK-vs.-TW and HCK-vs.-HW comparisons, the DEGs were assigned to 48 functional terms in the biological process, cellular component, and molecular function categories (Figure 8). Among these functional terms, “metabolic process” and “cellular process” were the most predominant classes in the biological process category for both TCK-vs.-TW and HCK-vs.-HW comparisons. For the cellular components category, the predominant classes of both TCK-vs.-TW and HCK-vs.-HW comparisons were the “cell”, “cell part” and “membrane”. However, the “cell” (1586 unigenes) and “cell part” (1586 unigenes) were the most dominant classes, and “membrane” was less predominant in the TCK-vs.-TW comparison, while for the HCK-vs.-HW comparison, the most dominant class was “membrane” (1901 unigenes), followed by “cell” (1848 unigenes) and “cell part” (1848 unigenes). In the molecular function category, the dominant classes were “catalytic activity” and “binding” for both the TCK-vs.-TW and HCK-vs.-HW comparisons.

We then performed a KEGG pathway enrichment analysis of those DEGs to identify their potential biological pathways (Appendix A). Among the 20 topmost enriched KEGG pathways, protein processing in the endoplasmic reticulum (ko04141), carbon metabolism (ko01200), and glycolysis/gluconeogenesis (ko00010) were the predominant enriched pathways in *M. toringoides*, while protein processing in the endoplasmic reticulum (ko04141), the biosynthesis of amino acids (ko01230), and plant hormone signal transduction (ko04075) were the most enriched pathways in *M. hupehensis* after 3 days of waterlogging stress. In addition, we found that the flavonoid biosynthesis (ko00941) pathway accounted for 1 of the 20 topmost enriched pathways in *M. hupehensis* under waterlogging stress, while it was not present in *M. toringoides*.

### 2.9. Expression Analysis of Flavonoid-Related DEGs and Hormone-Related DEGs

Since the KEGG analysis indicated some of the DEGs that were highly enriched in the pathways of flavonoid biosynthesis (ko00941) and plant hormone signal transduction (ko04075) (Appendix A), we paid particular attention to the flavonoid-related DEGs and hormone-related DEGs in *M. hupehensis* under waterlogging stress. As shown in Figure 9, several key genes that are involved in the biosynthesis of flavonoids were highly expressed in *M. hupehensis*, more than in *M. toringoides* under waterlogging stress, including genes encoding chalcone synthase (MD00G113200, MD04G1003300, MD13G1285100, MD04G1003000, and MD04G1003400), leucoanthocyanidin reductase (MD16G1048500, MD13G1046900, and MD06G1211400), flavanone 4-reductase (MD08G1028600), flavanone 3′-hydroxylase (MD14G1210700), flavonoid 3 beta-hydroxylase (MD02G1132200), flavonol synthase (MD15G1353800), dihydroflavonol 4-reductase (MD15G1024100), anthocyanidin reductase (MD05G1335600, MD10G1311100), and leucoanthocyanidin dioxygenase (MD03G1001100 and MD06G1071600). On the contrary, waterlogging stress decreased the expression of MD01G1118000 and MD01G1118100 genes encoding chalcone-flavonone isomerase in both *M. toringoides* and *M. hupehensis*, and the gene MD01G1118000 was more downregulated in *M. toringoides* compared with *M. hupehensis.* The differences in the expression of flavonoid-related genes between *M. toringoides* and *M. hupehensis* suggested a different role of flavonoids in *M. toringoides* and *M. hupehensis* in response to waterlogging stress.

Furthermore, we also identified four ethylene-related DEGs between *M. toringoides* and *M. hupehensis* in response to waterlogging stress (Figure 10). After waterlogging stress, although MD16G1212500 was annotated as an ethylene receptor, it was downregulated in both *M. toringoides* and *M. hupehensis*; MD13G1209700, annotated as an ethylene receptor, and MD03G1292200, annotated as a probable ethylene-response sensor, were significantly downregulated; and MD11G1306200, annotated as an ethylene-response sensor, was upregulated in *M. hupehensis*; meanwhile, they were all unchanged in *M. toringoides* (Figure 10). Furthermore, we also identified 25 auxin-related DEGs, including 7 auxin-transport-like proteins, 8 auxin-induced proteins, 5 auxin-responsive proteins, and 5 auxin-response factors (Figure 10). For the auxin-transport-like proteins, MD05G1118600, MD12G1162400, MD04G1149300, MD10G1121700, MD15G1355600, MD08G1169200, and MD07G1215900 were all upregulated in *M. hupehensis* after waterlogging stress, while they remained with no changes in *M. toringoides*. Auxin-induced proteins were also differently regulated between *M. toringoides* and *M. hupehensis* under waterlogging stress: except for the two auxin-induced proteins (MD10G1059800 and MD10G1060800) that were downregulated, the other six auxin-induced proteins (MD05G1205800, MD13G1204700, MD05G1052000, MD10G1192900, MD10G1061300, and MD10G1060700) were all upregulated in *M. hupehensis*, while most auxin-induced proteins remained unaffected by waterlogging stress in *M. toringoides* apart from MD05G1205800 and MD13G1204700, which were upregulated, and MD10G1060700, which was downregulated. For the auxin-responsive proteins, MD12G1241700 and MD15G1391700 were downregulated in *M. hupehensis*, but upregulated in *M. toringoides*. On the contrary, MD15G1191800 was upregulated in *M. hupehensis*, but downregulated in *M. toringoides*. Apart from those, MD13G1205000 and MD07G1297400 were both highly induced in *M. toringoides* and *M. hupehensis* by waterlogging stress. Among the auxin-response factors, after waterlogging stress, both MD01G1083400 and MD06G1111100 were upregulated in *M. toringoides* and *M. hupehensis*. However, MD15G1014400 and MD15G1221400 were only upregulated in *M. hupehensis*, while they were not affected in *M. toringoides*. In addition, MD08G1015500 was upregulated in *M. toringoides*, but was downregulated in *M. hupehensis*. Apart from ethylene- and auxin-related DEGs, we also identified one JA-related DEG (MD17G1081000, annotated as jasmonic acid–amido synthetase) and two ABA-related DEGs (MD15G1060800 and MD04G1165000, annotated as abscisic acid receptors), which were highly induced in *M. hupehensis* but were unaffected in *M. toringoides* (Figure 10).

Taken together, the observed differentially expressed hormone-related DEGs indicated a potential role of hormones in the regulation of the waterlogging tolerance mechanisms of *M. toringoides* and *M. hupehensis.*

### 2.10. qRT-PCR Validation and Analysis of Hormone-Related Genes

Based on the expression analysis of the hormone-related genes, we selected the six candidate genes with the highest expressions and investigated the transcriptional level of these six unigenes via qRT-PCR in *M. toringoides* and *M. hupehensis* under waterlogging stress, including abscisic acid receptor (MD15G1060800), jasmonic acid–amido synthetase (MD17G1081000), ethylene-response sensor (MD11G1306200), auxin-transport-like protein (MD10G1121700), auxin-induced protein (MD10G1061300), and auxin-responsive factor (MD01G1083400). As shown in Figure 11, all of the selected six unigenes were more highly induced in *M. hupehensis* compared with *M. toringoides* during the period of waterlogging stress, as per the results of the Illumina HiSeq sequencing (Figure 10). Taken together, the qRT-PCR results indicated that these six hormone-related genes might play an important role in regulating the waterlogging tolerance of *M. hupehensis*, and the consistency of the qRT-PCR results with the RNA-seq experiment further validated the reliability of our RNA-seq data.

## 3. Discussion

Waterlogging stress often causes leaf chlorosis, such as leaf wilting, yellowing, and even cellular death [23,35,44]. This was indeed observed in the present work, where *M. toringoides* showed wilted and yellowing leaves under waterlogging stress, and the leaves were even more withered and dead after prolonged waterlogging stress (Figure 1). However, these phenotypical changes were not observed in the apple rootstock *M. hupehensis*, suggesting that *M. hupehensis* was capable of maintaining normal growth under waterlogging stress (Figure 1). This observation further validated the hypothesis that *M. hupehensis* was more tolerant than *M. toringoides* to waterlogging stress.

### 3.1. Physiological and Morphological Changes under Waterlogging Stress

One of the primary effects of waterlogging stress is the depletion of O_2_. An insufficient O_2_ supply often results in an increase in ROS accumulation, such as O_2_^•−^ and H_2_O_2_, leading to lipid peroxidation and damage to the cell membrane, eventually causing leaf senescence [5,8,23]. O_2_^•−^ and H_2_O_2_ accumulations are commonly reflected by NBT and DAB staining, and damage to the cell membrane is often reflected by the measurement of electrolyte leakage (EL), a common parameter representing the integrity of the cell membrane and often associated with ROS accumulation [45,46]. By adopting these methodologies, we found that waterlogging stress induced severe damage to the cell membrane and a more significant increase in O_2_^•−^ and H_2_O_2_ in *M. toringoides* than in *M. hupehensis*, as observed by electrolyte leakage (EL) and NBT and DAB staining, respectively (Figure 2A,B and Figure 3). Based on these results, we speculated that the differences in ROS accumulations accounted for the distinct tolerance between *M. hupehensis* and *M. toringoides*. The waterlogging-tolerant *M. hupehensis* was able to keep the ROS accumulation to a slightly lower level to avoid or release the damage to the cell membrane, while the waterlogging-sensitive *M. toringoides* displayed higher ROS accumulations that led to more severe damage to the cell membrane.

The other difference between *M. hupehensis* and *M. toringoides* was correlated with photosynthesis under waterlogging stress. Waterlogging stress can induce stomata closure, which prevents gas exchange and leads to insufficient CO_2_ absorption; thus, the photosynthetic rate is affected [5,43,47,48,49]. In the present study, the stomata size and density were significantly decreased in *M. toringoides* by waterlogging stress, while the stomata could be maintained in its original size and density in *M. hupehensis* (Figure 4). This indicated that *M. hupehensis* leaves better facilitated the stomata opening and performed better gas exchange and CO_2_ acquisition under waterlogging stress. Furthermore, since *Fv*/*Fm* represents the photochemistry of photosynthesis, waterlogging induced a stronger reduction in *Fv*/*Fm* in *M. toringoides* than in *M. hupehensis*, indicating a stronger injury to the photochemistry of photosynthesis in *M. toringoides* than in *M. hupehensis* (Figure 5). These results were similar to those in previous reports, where the waterlogging-sensitive apple *Hongro* and evergreen tree *Pouteria glomerata* showed a significant reduction in *Fv*/*Fm* and photosynthesis than the waterlogging-tolerant apple *Fuji* and evergreen tree *Cecropia latiloba*, respectively [32,50]. Thus, we speculated that the tolerant *M. hupehensis* leaves had the ability to control stomata functions and maintain their normal photochemical process, ensuring the better photosynthesis required for normal growth and development under waterlogging stress.

Waterlogging stress induces biosynthesis and the accumulations of ethylene. Depending on the plant species, growth phase, and stress durations, ethylene is synthesized to either enhance the stress responses required for survival or accelerate the stress-induced symptoms effects [51]. In the present work, we observed different behaviors in the ethylene-responses of *M. toringoides* and *M. hupehensis* under waterlogging stress: in *M. hupehensis*, ethylene production kept gradually increasing during the waterlogging stress for 48 h, while in *M. toringoides*, ethylene production sharply increased at the initial 9 h mark after waterlogging stress and then declined, followed by a second ethylene peak when waterlogging stress was prolonged to 48 h (Figure 6). This observation in *M. toringoides* clearly fitted well with the two-phase ethylene-response model [52]: a transient increase in ethylene in the few hours after stress treatment acts as a signaling compound that activates the transcriptional responses and induces stress resistance, thus providing a protective process. When the stress is prolonged, a second increment in ethylene is often observed, which no longer acts as a signaling compound but serves as a toxic compound, inducing leaf senescence or chlorosis. According to this model, in *M. toringoides*, the initial ethylene peak at 9 h was interpreted as a protective response (signal) that activated transcriptional responses, while the second peak at 48 h after waterlogging stress (Figure 6) was interpreted as a deleterious process that induced leaf senescence and chlorosis, which was observed in *M. toringoides* (Figure 1). In comparison, the gradual increase in lower ethylene production in response to waterlogging stress in *M. hupehensis* always seemed to act as a protective response and as a signaling compound. Therefore, we assumed that in the waterlogging-tolerant *M. hupenensis*, waterlogging-induced ethylene acts more as a signaling compound and not as a stress hormone, whereas dual action is suspected in the waterlogging-sensitive *M. toringoides.*

### 3.2. Differentially Expressed Genes Involved in Flavonoid Biosynthesis under Waterlogging Stress

Flavonoids are one of the important plant secondary metabolisms and are mainly composed of six subgroups, namely, chalcones, flavones, flavonols, flavandiols, anthocyanins, and proanthocyanidins or condensed tannins, which are catalyzed by multiple key enzymes, respectively [16,53]. There is increasing evidence that flavonoids have multiple functions in response to various stresses, such as UV radiation, pathogen infection, and auxin transport [16,54]. However, until now, less is known about whether flavonoids are involved in regulating waterlogging stress tolerance. In this study, we identified several flavonoid-related DEGs in response to waterlogging stress. Interestingly, except for two genes encoding chalcone-flavonone isomerase (MD01G1118000 and MD01G1118100)—which was downregulated by waterlogging stress—under waterlogging stress, almost all flavonoid-related DEGs were induced in significantly higher numbers in the tolerant *M. hupehensis* than in *M. toringoides*, including genes encoding chalcone synthase (MD00G1132100, MD04G1003300, MD13G1285100, MD04G1003000, and MD04G1003400), leucoanthocyanidin reductase (MD16G1048500, MD13G1046900, and MD06G1211400), flavanone 4-reductase (MD08G1028600), flavanone 3′-hydroxylase (MD14G1210700), flavonoid 3 beta-hydroxylase (MD02G1132200), flavonol synthase (MD15G1353800), dihydroflavonol 4-reductase (MD15G1024100), anthocyanidin reductase (MD05G1335600 and MD10G1311100), and leucoanthocyanidin dioxygenase (MD03G1001100 and MD06G1071600) (Figure 9). This observation indicates a possible role of flavonoids in the regulation of waterlogging tolerance, and upregulated flavonoid-related DEGs may contribute to the higher resistance of *M. hupehensis* to waterlogging stress. Furthermore, it has been reported that flavonoids act as antioxidants due to their polyphenolic structures, and that they have the ability to restrict membrane fluidity, limiting the diffusion of free radicals and restricting peroxidative reactions [55,56]. Together with the observation of fewer accumulations of O_2_^•−^ and H_2_O_2_ in the tolerant *M. hupehensis* under waterlogging stress (Figure 3), our data further suggested a possible link between upregulated flavonoid-related genes and the free radical scavenging mechanism, which together probably modulate the waterlogging tolerance of *M. hupehensis*. However, further experiments are required to explore this aspect.

### 3.3. Differentially Expressed Genes Involved in Hormone Biosynthesis and Signaling under Waterlogging Stress

Plant hormones are vital endogenous signals that mediate complex signaling in response to waterlogging stress [5]. Among the plant hormones, ethylene has been reported to act as a primary signal in response to waterlogging stress [5,11,36,37]. Here, we identified four DEGs involved in ethylene signaling (Figure 10). Except for the ethylene receptor (MD16G1212500) that was downregulated in *M. toringoides* and remained unchanged in *M. hupehensis*, the ethylene receptor (MD13G1209700), the probable ethylene-response sensor (MD03G1292200), and the ethylene-response sensor (MD11G1306200) were either downregulated or upregulated in *M. hupehensis*, respectively, while they remained unchanged in *M. toringoides* (Figure 10). The greater number of changes in the expression of these ethylene-signaling-related DEGs identified in *M. hupehensis* might suggest it has a stronger ethylene signaling to waterlogging stress than *M. toringoides* does, which is consistent with our observation and speculation that the gradual increase in ethylene in *M. hupehensis* acts as signaling compound but not a stress hormone (Figure 6). However, the functional investigation of these four DEGs is further needed and may help reveal the possible link of ethylene signaling and the tolerance mechanism of *M. hupehensis*.

Auxin transport and signalling also play important roles in regulating plant resistance to waterlogging stress [57,58]. Ethylene also can promote the accumulation and transport of auxin to induce adventitious root initiation under waterlogging stress [5,58,59]. In a comparative transcriptome study of the two different waterlogging-tolerant Chrysanthemum morifolium cultivars, genes related to auxin signaling and transport are differentially expressed between waterlogging-tolerant “*Nannongxuefeng*” than in waterlogging-sensitive “*Qinglu*”. For instance, auxin-response factor (ARF7) and ARF19-like and auxin-transporter-like protein 2 are more induced in waterlogging-tolerant “*Nannongxuefeng*” than in waterlogging-sensitive “*Qinglu*”, while ARF7-like and ARF5 are found to be more induced in “*Qinglu*” by waterlogging [60].In the present work, we also identified 25 auxin-related DEGs between *M. hupehensis* and *M. toringoides* under waterlogging stress (Figure 10). Notably, all seven identified auxin-transport-like proteins (MD05G1118600, MD12G1162400, MD04G1149300, MD10G1121700, MD15G1355600, MD08G1169200, and MD07G1215900) were more induced by waterlogging stress in *M. hupehensis* than in *M. toringoides*, suggesting that auxin-transport-like proteins play fundamental roles in regulating the tolerance of *M. hupehensis* to waterlogging stress. In addition, four auxin-induced proteins (MD05G1052000, MD10G1192900, MD10G1061300, and MD10G1060700), the auxin-responsive proteins (MD15G1191800), and two auxin-response factors (MD15G1014400 and MD15G1221400) were more expressed in *M. hupehensis* than in *M. toringoides* under waterlogging stress, while two auxin-responsive proteins (MD12G1241700 and MD15G1391700) and one auxin-response factor (MD08G1015500) were more suppressed by waterlogging stress in *M. toringoides*, indicating these auxin-related DEGs may play different roles in regulating the tolerance of *M. hupehensis* to waterlogging stress.

Jasmonic acid (JA) and abscisic acid (ABA) are also key regulators of waterlogging stress [5]. However, the regulation of JA and ABA in waterlogging tolerance remains unclear, as opposite results have been observed. For instance, in a comparative study of two different waterlogging-tolerant cucumbers, JA content was found to be significantly decreased in the waterlogging-tolerant cucumber *Zaoer-N*, but was increased in the waterlogging-sensitive cucumber *Pepino*, suggesting a negative role of JA in regulating waterlogging stress [61], while the exogenous application of JA on soybean was reported to release the damage of waterlogging stress [62]. ABA has been reported to negatively regulate waterlogging stress tolerance since ABA content is more significantly decreased by waterlogging stress in waterlogging-tolerant soybean than in waterlogging-sensitive soybean [63]; however, in Arabidopsis, the overexpression of RAP2.6L increases ABA biosynthesis and increases ABA content, promoting stronger antioxidant enzyme activities, inducing stomata closure, and delaying leaf senescence under waterlogging stress [64]. Therefore, it seems that the observed differences in the regulatory function of JA and ABA in plant waterlogging tolerance might highly depend on the species used in the experiments. In the present work, we found that jasmonic acid–amido synthetase (MD17G1081000) and two abscisic acid receptors (MD15G1060800 and MD04G1165000) were more highly induced in the waterlogging-tolerant *M. hupehensis*, but were unaffected in the waterlogging-sensitive *M. toringoides* (Figure 10), suggesting that JA biosynthesis and ABA signaling are positively involved in regulating the tolerance of *M. hupehensis* to waterlogging stress. However, their precise functions should be further elucidated.

## 4. Materials and Methods

### 4.1. Plant Materials and Cultivation

The seeds of *M. toringoides* and *M. hupehensis* were stratified in sand at 0 to 4 °C in a refrigerator from the end of December 2019 to March 2020 at the experimental base of Henan Agricultural University. After germination, seeds with uniform size at the same stage of germination were selected and sowed into 32-hole seedling trays (54 mm × 28 mm × 50 mm) that contained gravel and seedling substrate (1:1). When they developed 6 to 8 true leaves, the healthy seedlings with uniform size were transplanted into plastic pots (18 cm × 12.5 cm × 15 cm per pot, with one seedling) that contained a mixture of field soil, seedling substrate, and chicken manure (3:2:1). After that, the seedlings were watered normally every 3–5 days to sustain normal growth until waterlogging treatment.

### 4.2. Waterlogging Treatment

Waterlogging treatment was conducted in July 2020. The seedlings of each genotype with uniform size were randomly selected and classified into control group and waterlogging treatment group. Each treatment contained three biological replicates with 60 plants in each replicate, resulting in a total of 180 plants per treatment (Appendix A). For the control group, the growth of seedlings was kept in basins at a constant 60% relative water content and irrigated and replenished daily using a HH2 Moisture Meter (Delta-T Devices Ltd., Cambridge, UK), while for the waterlogging treatment, the pots were fully immersed in water that reached 3 cm above the soil surface. The morphological changes were recorded following 0, 3, and 6 days of waterlogging stress. A summary of the number of seedlings sampled for the below experiments is shown in Appendix A.

### 4.3. Measurement of Electrolyte Leakage (EL) and Leaf Relative Water Content

The 2~3 leaves from the apical part of each genotype were harvested before and after waterlogging treatment for 5 consecutive days to measure the electrolyte leakage (EL) and leaf relative water content. The EL and leaf relative water content were determined as described in [20].

### 4.4. Detection of O_2_^•−^ and H_2_O_2_ Accumulation

A total of 3~6 fully expanded leaves from the middle and upper part of each genotype were harvested after 3 days of waterlogging treatment, and the untreated leaves were also collected as control. The O_2_^•−^ and H_2_O_2_ accumulation were detected by nitro blue tetrazolium (NBT) and diaminobenzidine (DAB) staining, respectively, according to [43]. Briefly, leaf samples were immersed in either the freshly prepared NBT staining solution (1 mg/mL NBT in 10 mM HEPES (pH 7.5) with 0.1% Triton X-100) or the fresh DAB staining solution (1 mg/mL DAB in 10 mM sodium phosphate buffer (pH 7.5) with 0.1% Triton X-100). After incubation in darkness for almost 12 h, the staining solutions were removed until brown or blue spots were visible in the leaves. Then, the leaf samples were immersed in 95% ethanol and heated in a 95 °C water bath for 15 min to remove the chlorophyll. Then, the leaf samples were preserved in 95% ethanol until photographing.

### 4.5. Stomata Observation and Analysis

After 3 days of waterlogging treatment, the 5th to 8th leaves from the apical of control and waterlogging-treated *M. toringoides* and *M. hupehensis* were collected. The stomata were observed according to [46]. Briefly, the leaf samples were cut into 2~3 cm fragments and immediately fixed in 4% glutaraldehyde with a vacuum pump for 30 min. After fixation for 6 h, the leaf samples were rinsed with 0.2 M PBS buffer 5 times. After dehydration with a series of ethanol treatments (30%, 50%, and 70% for 15 min, 80%, 90%, and 100% for 30 min), the leaf samples were rinsed twice with isoamyl acetate for 25 min each and dried with a Hitachi HCP-2 zero-boundary-point dryer (Tokyo, Japan). Then, the stomata were observed and imaged via SEM. The width and length of stomata were calculated using Image-pro plus 6.0 (Media Cybernetics, Inc., Rockville, MD, USA). For the density of stomata, the number of stomata in 6 fields was counted, and the density of stomata was calculated using the number of stomata in 6 fields divided by the area of the field.

### 4.6. Measurement of Ethylene Production

Six control and waterlogged seedlings of *M. toringoides* and *M. hupehensis* with uniform size were randomly selected and immediately transferred into a closed plastic box at room temperature, 25 °C. After transferring at time points of 0 h, 3 h, 6 h, 9 h, 12 h, 24 h, 48 h, 96 h, and 144 h, 1000 μL of the headspace air was extracted from the closed plastic box with a syringe and injected into a GC2010 Plus gas chromatography device (Shimadz, Kyoto, Japan) equipped with an auto sampler and a flame ionization detector using a capillary GC column (Zebron ZB-WAX plus, 30 m, 0.25 mm ID, 0.25 µm). Helium was used as carrier and make-up gas. The injection port and detector temperatures were 240 °C and 250 °C, respectively. Then, the ethylene production was determined as described in [60], and each treatment contained at least three independent replicates.

### 4.7. Measurement of the Leaf Maximal Photochemical Efficiency (Fv/Fm)

For measuring the leaf maximal photochemical efficiency (*Fv*/*Fm*), after 3 days of waterlogging stress, the control and waterlogged seedlings of *M. toringoides* and *M. hupehensis* with uniform size were randomly selected and placed in darkness for 2 h. *Fv*/*Fm* was measured using a Chlorophyll Fluorescence Meter (PAM-2500). Three biological replicates for each treatment were performed.

### 4.8. RNA Extraction, cDNA Library Construction, and Sequencing

The leaves from controlled and waterlogged *M. toringoides* and *M. hupehensis* were harvested after 3 days of waterlogging stress and immediately immersed in liquid nitrogen and stored in −80 °C until RNA extraction. The extraction of total RNA was conducted according to the manufacturer’s instructions of Plant Total RNA Isolation Kit (Sangon Biotech Co., Ltd., Shanghai, China). After verification of RNA quality and concentration by 1% agarose gel and an ND-1000 Spectrophotometer (NanoDrop, Wilmington, DE, USA), only high-quality RNA was used in subsequent experiments. The subsequent libraries were constructed following the RNA-seq library constructed flow path and were sequenced with an Illumina HiSeq 4000 by the Biomarker Biotechnology Corporation (Beijing, China). After removing the low-quality reads, the clean reads were mapped to the apple reference genome (GDDH13 Version 1.1) [65] using the TopHat 2 and Bowtie 2 program.

### 4.9. Identification of the DEGs

The gene expression level was estimated using the methods of fragments per kilobase of transcript sequence per million (FPKM) [66]. The differentially expressed genes (DEGs) of the control and waterlogged *M. toringoides* and *M. hupehensis* at the same time point were identified with the DESeq2 R package [67]. Only the genes with both FDR (false discovery rate) < 0.05 and log2 (fold change) ≥ 1 between two samples were considered DEGs [68]. Gene ontology (GO) enrichment analysis of DEGs was implemented using the GOseq R packages according to a Wallenius non-central hyper-geometric distribution [69]. In addition, the enriched metabolic pathways of the DEGs were identified using the Kyoto Encyclopedia of Genes and Genomes (KEGG, http://www.genome.jp/kegg/, accessed on 15 October 2021), through the KOBAS program.

### 4.10. qRT-PCR Validation and Analysis

The six candidate genes of hormone-related DEGs with the highest expression in RNA-seq were selected and investigated by quantitative real-time PCR (qRT-PCR) to validate the RNA-seq experiment. The specific primers of those six genes are listed in Appendix A. qRT-PCR was performed according to [42]. The housekeeping gene Actin was employed as normalization, and the relative expression of each gene was calculated using the 2^−ΔΔCt^ method according to [70].

## 5. Conclusions

Our results demonstrate that *M. hupehensis* was more tolerant than *M. toringoides* to waterlogging stress, as observed by less leaf chlorosis. *M. hupehensis* showed the ability to maintain normal O_2_ and H_2_O_2_ accumulation and prevent lipid oxidation damage and to improve the stomatal opening and *Fv*/*Fm* to maintain normal photosynthesis under waterlogging stress. Notably, the continuous ethylene signaling contributed to another tolerance mechanism of *M. hupehensis* to waterlogging stress. The difference in DEGs related to flavonoids and hormone signaling between *M. hupehensis* and *M. toringoides* also suggested a possible role of the biosynthesis of flavonoids and hormone signaling in the regulation of waterlogging stress tolerance. Based on these results, we could propose a hypothetical model of the waterlogging tolerance mechanism of apple (Figure 12), which helps us better understand the underlying mechanism of apples against waterlogging stress.

## Figures and Tables

**Figure 1 ijms-24-09298-f001:**
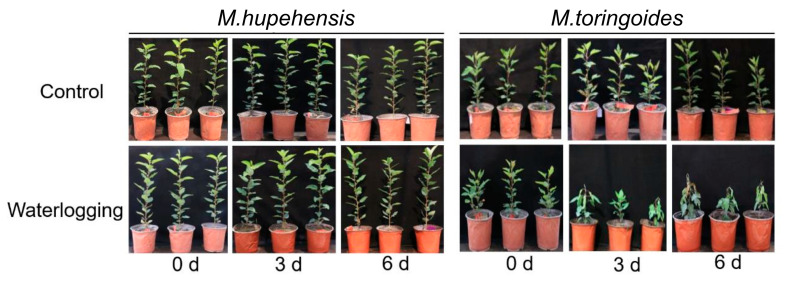
Representative images of phenotypical changes in *M. toringoides* and *M. hupehensis* after 0, 3, and 6 days of waterlogging treatment.

**Figure 2 ijms-24-09298-f002:**
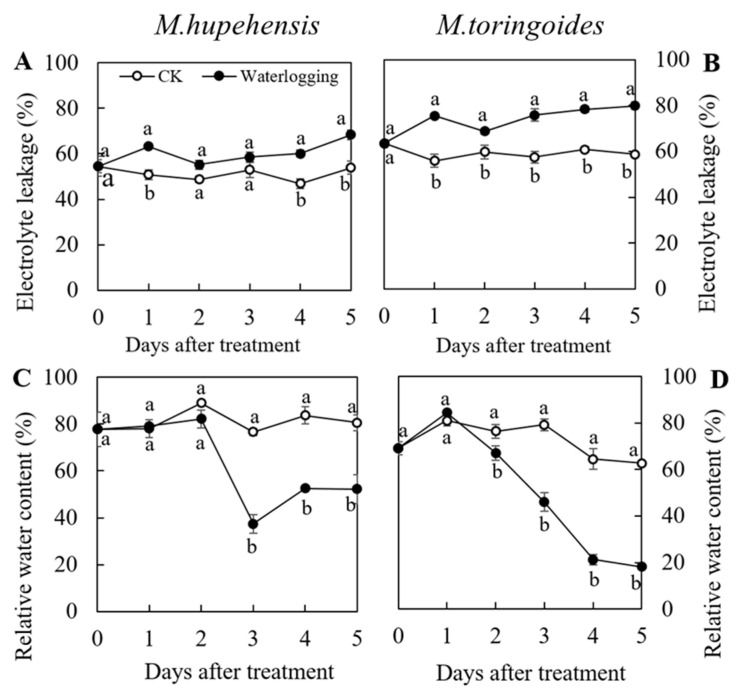
Electrolyte leakage and leaf relative water content in *M. toringoides* and *M. hupehensis* under waterlogging stress. (**A**,**B**), electrolyte leakage of *M. hupehensis* and *M. toringoides* after 5 consecutive days of waterlogging stress, respectively. (**C**,**D**), leaf relative water content of *M. hupehensis* and *M. toringoides* after 5 consecutive days of waterlogging stress, respectively. The white and black dots represent data analyzed from control (CK) and waterlogging treatment (waterlogging) of *M. toringoides* and *M. hupehensis*, respectively. Data are mean values ± standard errors (n = 3). Different letters indicate significant differences between the control and treated plants in each species at *p* < 0.05 by LSD’s test.

**Figure 3 ijms-24-09298-f003:**
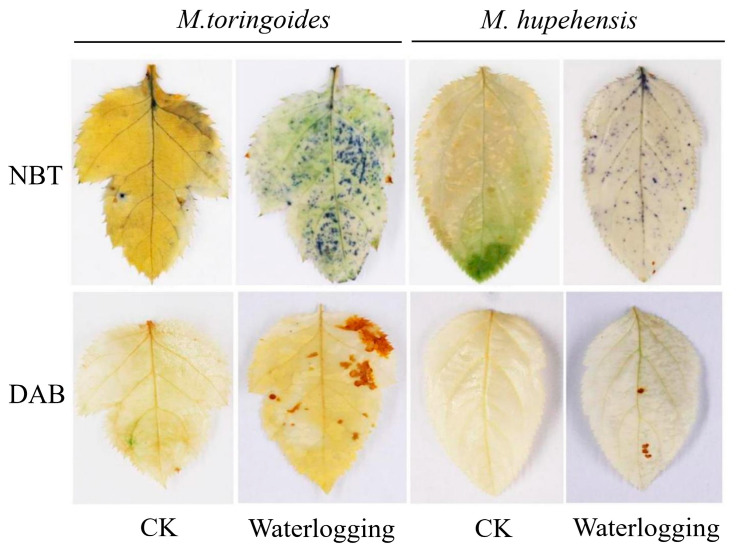
O_2_^•−^ and H_2_O_2_ accumulation in *M. toringoides* and *M. hupehensis* under waterlogging stress for 3 days. O_2_^•−^ and H_2_O_2_ production were detected using NBT and DAB. CK: control of *M. toringoides* and *M. hupehensis*; Waterlogging: waterlogging-treated *M. toringoides* and *M. hupehensis*.

**Figure 4 ijms-24-09298-f004:**
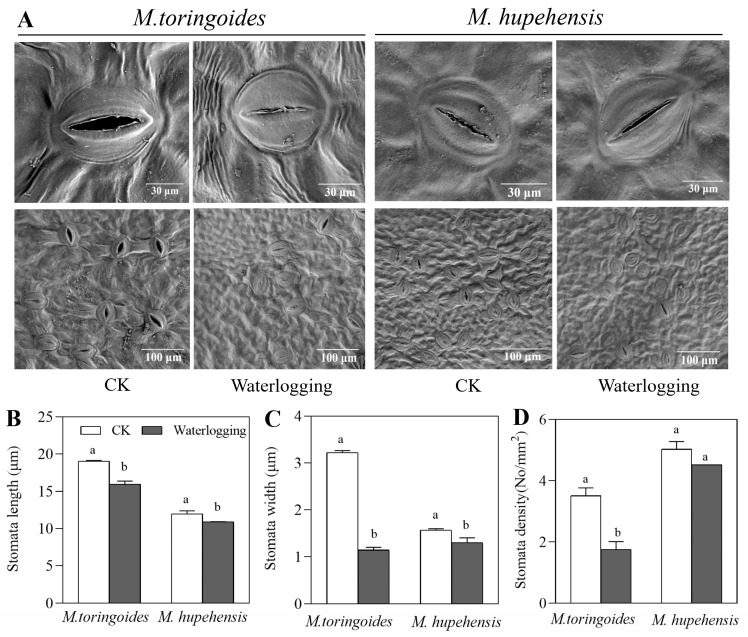
Stomatal characteristics in leaves of *M. toringoides* and *M. hupehensis* under waterlogging stress for 3 days. (**A**) Images from SEM of stomata, (**B**) stomata lengths, (**C**) stomata widths, and (**D**) stomata density. CK: control of *M. toringoides* and *M. hupehensis*; Waterlogging: waterlogging-treated *M. toringoides* and *M. hupehensis*. Data are means ± standard errors of 20 images. Values not followed by same letters indicate significant differences between control and stressed plants, and LSD test was used (*p* < 0.05) to compare significant treatment means.

**Figure 5 ijms-24-09298-f005:**
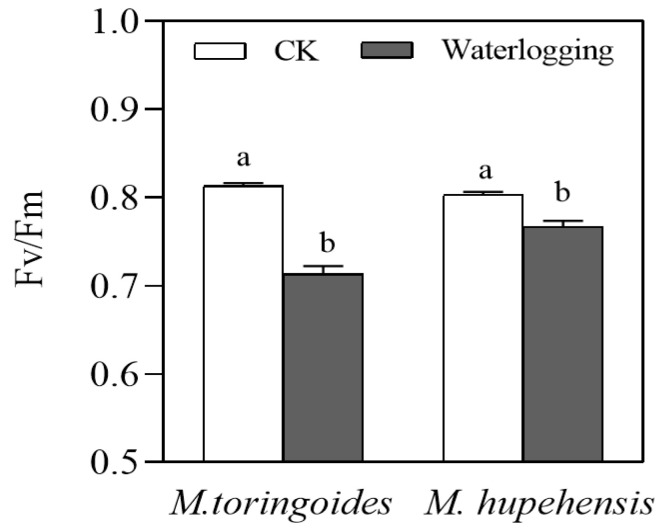
*Fv*/*Fm* in leaves of *M. toringoides* and *M. hupehensis* under waterlogging stress for 3 days. Data are mean values ± standard errors of three biological replicates and each biological replicate contains 6 independent determinations. a and b indicate significant differences between control (CK) and stressed (waterlogging) plants, and LSD test was used (*p* < 0.05) to compare significant treatments means.

**Figure 6 ijms-24-09298-f006:**
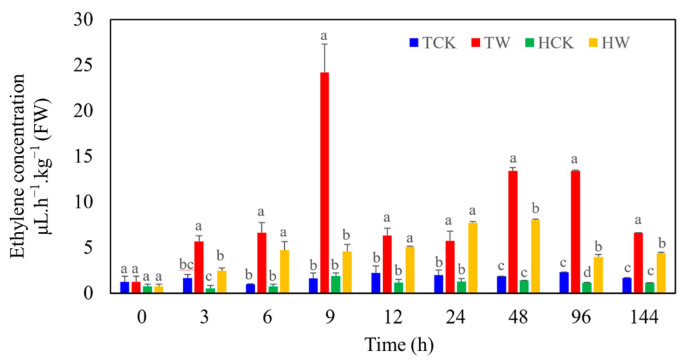
Ethylene production in *M. toringoides* and *M. hupehensis* under waterlogging stress. TCK: control of *M. toringoides;* TW: waterlogged *M. toringoides;* HCK: control of *M. hupehensis;* HW: waterlogged *M. hupehensis.* Data are mean values ± standard errors of 3 biological replicates, and each biological replicate contains 6 independent determinations. a, b and c represent the statistical differences between control and waterlogged *M. toringoides* and *M. hupehensis*. Statistical differences were determined using ANOVA, and LSD test was used (*p* < 0.05) to compare significant treatments means.

**Figure 7 ijms-24-09298-f007:**
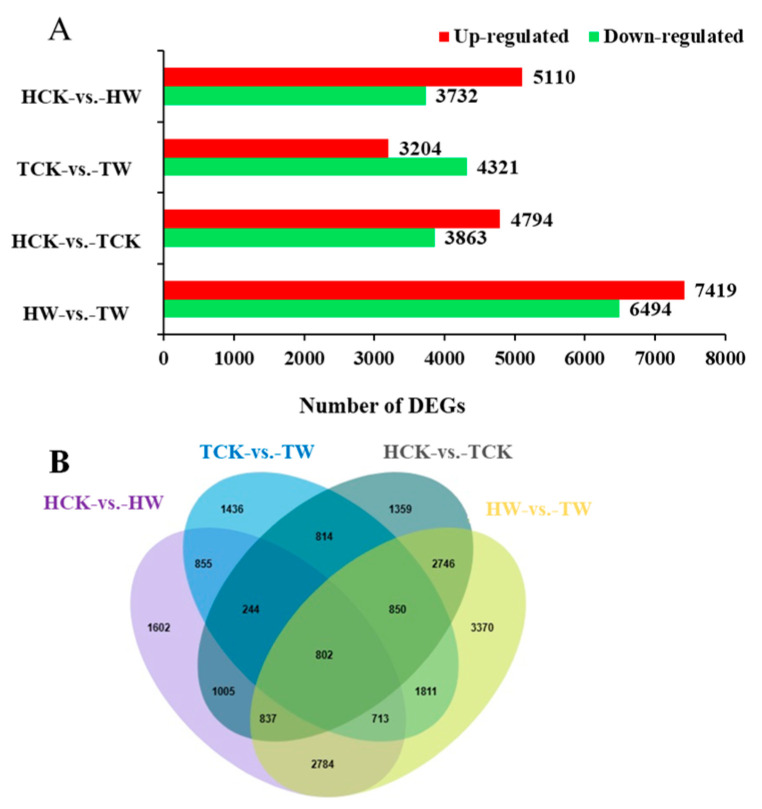
(**A**) Statistical analysis of upregulated and downregulated differentially expressed genes (DEGs) in *M. toringoides* and *M. hupehensis* in response to waterlogging stress. Red and green bars are the number of up- and downregulated genes in the comparison of HCK-vs.-HW, TCK-vs.-TW, HCK-vs.-TCK, and HW-vs.-TW, respectively. (**B**) Venn diagrams showing the number of up- and downregulated DEGs in the comparisons of HCK-vs.-HW, TCK-vs.-TW, HCK-vs.-TCK, and HW-vs.-TW, respectively. The overlapping regions showed the common DEGs among the four comparisons. TCK: control of *M. toringoides*; TW: waterlogged *M. toringoides*; HCK: control of *M. hupehensis*; HW: waterlogged *M. hupehensis*.

**Figure 8 ijms-24-09298-f008:**
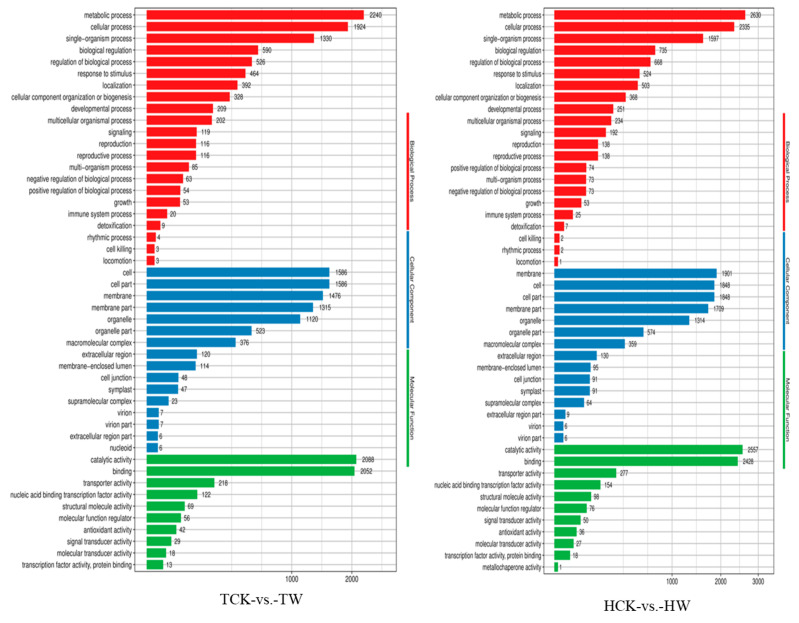
GO classification of unigenes in *M. toringoides* and *M. hupehensis* under waterlogging stress. Results are summarized in three main GO categories: biological processes, cellular components, and molecular functions.

**Figure 9 ijms-24-09298-f009:**
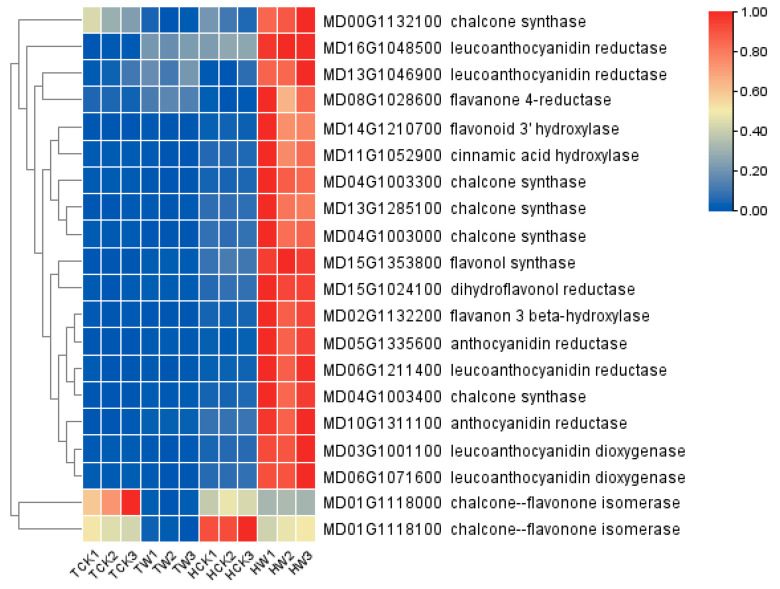
Heatmap analysis of flavonoid-related genes. The sample are displayed below each column. Gene ID and annotation of each gene are shown. The expressions of gene are displayed in different colors. Red means high expression, and blue means low expression.

**Figure 10 ijms-24-09298-f010:**
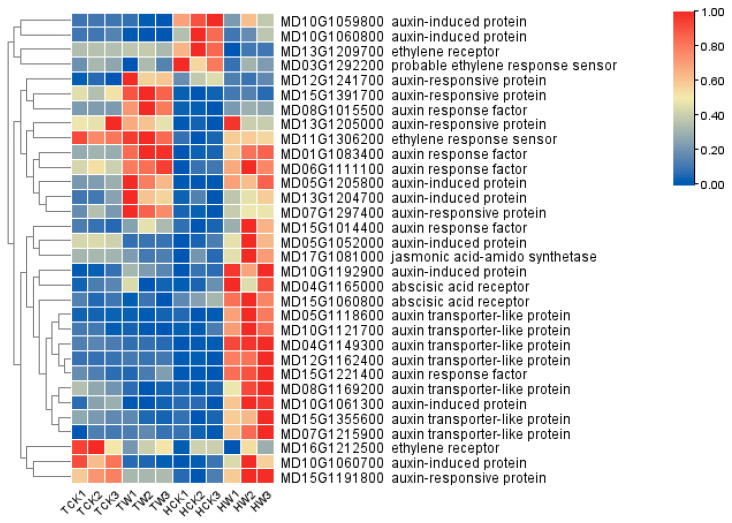
Heatmap analysis of hormone-related genes. The samples are displayed below each column. Gene ID and annotation of each gene are shown. The expressions of the genes are displayed in different colors. Red means high expression and blue means low expression.

**Figure 11 ijms-24-09298-f011:**
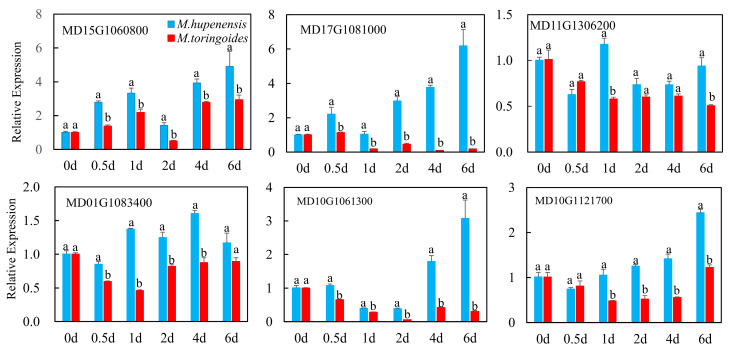
qRT-PCR validation of the differential expression of 6 genes involved in hormones in *M. toringoides* and *M. hupehensis* after 0 d, 0.5 d, 1 d, 2 d, 4 d, and 6 d of waterlogging stress. Vertical bars indicate standard deviation from three independent technical replicates. a and b indicate significant differences between the control and waterlogged plants in each species at *p* < 0.05 by LSD’s test.

**Figure 12 ijms-24-09298-f012:**
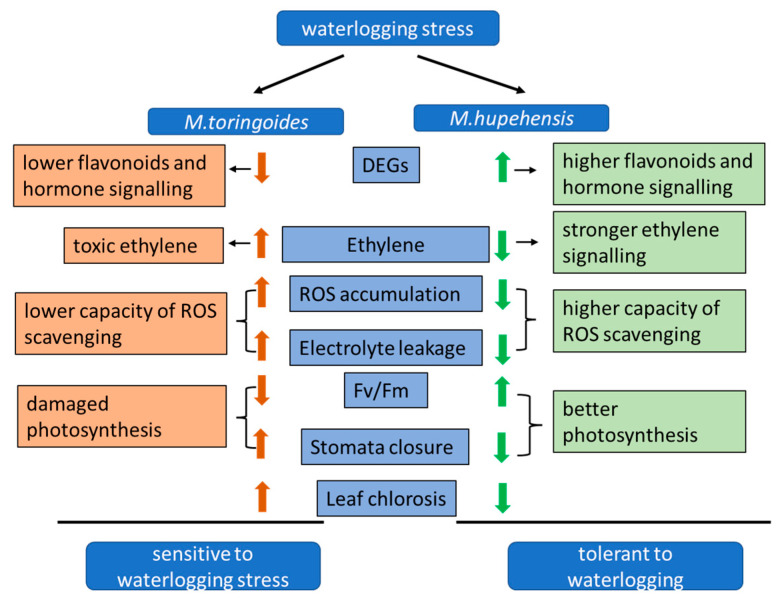
A hypothetical model of the response to waterlogging stress in apple rootstock by physiological and morphological changes and regulating gene expression. The arrow up or down means up regulated or downregulated, respectively.

## Data Availability

The data presented in this study are available on request from the corresponding author. The raw data of RNA-seq for this study had been deposited in the public database Sequence Read Archive (SRA) with bioproject: PRJNA950188.

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
