# Peer review of "Comparative Physiological and Transcriptome Analysis Reveals Potential Pathways and Specific Genes Involved in Waterlogging Tolerance in Apple Rootstocks"

_ijms, 2023, doi:10.3390/ijms24119298_

Round 1

Reviewer 1 Report

In the MS titled “Comparative physiological and transcriptome analysis reveals potential pathways and specific genes involved in waterlogging tolerance in apple rootstocks”, the authors investigated the effect of waterlogging on apple seedlings. This study is interesting given the impact of flooding on agriculture worldwide. After reading the manuscript, I noted some areas that require improvement and I have highlighted them either in form of questions or comments. Generally, the whole manuscript requires grammar improvement.

Materials and methods

4.1: How did you maintain 0-4oC temperature in the field? If the reason was to break dormancy, why were seeds not planted directly into the germination trays and kept at controlled temperature?

4.2: Experimental design is not clearly described. The concept of biological replication and blocking is quite confusing in the description. How did you maintain the 60% relative water content in the control treatment? When you state that the seedlings in the waterlogging treatment were fully immersed in water, this can easily be misinterpreted that the whole plant was submerged which is not true.  

4.3-4.4: The sampling strategy based on the experimental design is not very clear. You should show how the 720 plants were utilized in the different experiments (2 genotypes x 2 treatments x 3 blocks x 60 replicates). Figure S1 does not fully illustrate how the sampling was done for the six experiments conducted. An additional supplemental table would help clarify this.

4.8: “… the remaining clean reads were further analysed by BMK cloud server and mapped to the apple reference genome …”. This is a confusing statement about read processing.

Figure 12 is less intuitive for describing the hypothesis regarding waterlogging stress tolerance mechanisms.   

Results

The opening statement should indicate the kind of difference mechanisms you are referring to.

2.4: Check Figure 4C annotations. The statement on lines 198-199 is not substantiated by Figure 5 given a statistical significance.

Lines 223-224: Which two different roles does ethylene play in M. toringoides?

Figure 6: The abbreviations in the legend must be defined in the figure caption since it is the first time some are used.

2.6. The title “Transcriptome sequencing and read assembly” does not reflect the content of the paragraph. Any idea why more than 20% of the reads could not map to the reference?

Figure 7A legend is incomplete.

In section 2.8 and 2.9 there is a lot of redundancy between the text and figures. It is better to highlight the most important observations while adding a sound perspective and leave the details to the figures or tables.

Discussion

3.2. Title is not clear “Genes Involved in Flavonoids under waterlogging stress”

3.3. Title is not clear “Genes Involved in hormones under waterlogging stress”

There are so many grammatical errors throughout the manuscript. The manuscript could benefit from edits by a native English speaker or language editorial services. 

Author Response

Response to Reviewer 1

Thanks very much for your constructive comments, which were very valuable for improving our manuscript. We have read the comments and suggestions carefully and revised the manuscript accordingly. Point by point responses to your comments are listed below. Alog with other reviewers’ comments, we made a lot of changes in our revised manuscript, which we marked in red (changes mainly focusing on the reviewers’ comments) in the revised manuscript. In addition, we also asked Anita K Snyder, an English editor, to have our revised manuscript spell-checked and grammar-checked. We hope that the revised version of the manuscript is now acceptable for publication.

Point-by-point response:

Comment: 4.1: How did you maintain 0-4oC temperature in the field? If the reason was to break dormancy, why were seeds not planted directly into the germination trays and kept at controlled temperature?

Response: Thanks for your advice, however, to our experience, stratification in sand is a common way to break the dormancy of fruit trees' seeds. The winter temperature in is between -10℃and 5℃ in Zhengzhou, after years of testing, burying seeds in the sand the temperature can be maintained at 0 to 4℃, therefore, it is a good way to break seed dormancy. In addition, we've tried the author's suggestion before, but it didn't work out well. Put the seeds directly in the refrigerator, if there is too much water, the seeds will become moldy. If there is too little water, the seeds will not germinate.

Comment: 4.2: Experimental design is not clearly described. The concept of biological replication and blocking is quite confusing in the description. How did you maintain the 60% relative water content in the control treatment? When you state that the seedlings in the waterlogging treatment were fully immersed in water, this can easily be misinterpreted that the whole plant was submerged which is not true.  

Response: we could see the point that we make it confused. For the experimental design, we have 3 blocks that each block represents one biological replicate and each block (biological replicate) contains 60 independent plants. For each experiment, depending on the experiments, we sampled at least 3 plants from each block as one biological replicate, and in total 3 blocks/biological replicates. We have modified the supplemental Figure S1 and added a supplemental Table S4 to clarify it, as your suggestion in the next comment.

For the maintenance of 60% relative water content, a HH2 Moisture Meter (Delta-T Devices Ltd., Cambridge, UK) was adopt to measure daily, and then irrigated and replenished water daily accordingly.

For the third question, we are sorry for the misleading that it is not the seedling that were fully immersed in water, but the pots were fully immersed in water. We have changed the ‘seedlings’ to ‘pots’. Please see line 596-604.

Comment: 4.3-4.4: The sampling strategy based on the experimental design is not very clear. You should show how the 720 plants were utilized in the different experiments (2 genotypes x 2 treatments x 3 blocks x 60 replicates). Figure S1 does not fully illustrate how the sampling was done for the six experiments conducted. An additional supplemental table would help clarify this.

Response: We apologized for the confusing. As explained in the above comment, we used 3 blocks as 3 biological replicates. According to your suggestion, we have now added a new supplemental Table S4 to help clarify how many plants for each experiment used. Please see the supplemental Table S4. Especially for the ethylene production, we only took 6 independent seedling in each replicate to measure ethylene production during the whole time points, therefore, the total seedlings in the experiment of ethylene measurement is 6 independent seedlings in each replicate. We also rewrite the sentence in section 4.2, please see the changes in section 4.2.

Comment: 4.8: “… the remaining clean reads were further analysed by BMK cloud server and mapped to the apple reference genome …”. This is a confusing statement about read processing.

Response: we are sorry for the misleading, now we have rephased this sentence. Please see line 678-680 in the manuscript.

Comment: Figure 12 is less intuitive for describing the hypothesis regarding waterlogging stress tolerance mechanisms.   

Response: according to your question, we have made a new diagram of model for better understanding the changes between the two species under waterlogging stress, by clear showing the comparison of physiological changes (EL, Fv/Fm, ethylene, DEGs…) and leaf chlorosis… which we thought will be better enough. Please see the new Figure 12.

Results

Comment: The opening statement should indicate the kind of difference mechanisms you are referring to.

Response: Many thanks for your suggestion, we have revised the opening statement in each paragraph accordingly.

Comment: 2.4: Check Figure 4C annotations. The statement on lines 198-199 ‘This was indeed observed in the present study, where waterlogging treatment could induce the stomata closure of both M.toringoides and M.hupehensis (Figure 4A).’ is not substantiated by Figure 5 given a statistical significance.

Response: Many thanks for pointing out the issue, we have checked the Figure 4C and corrected it. For the second question, we do not think the two parameters should be substantiated by each other, rather that we used these two parameters to reveal different aspects of damaged photosynthesis and to compare the level of damage by waterlogging stress. In figure 5, we observed a decrease of Fv/Fm in both rootstocks, while waterlogging stress induced greater decrease in the sensitive M.toringoides than the tolerant M.hupehensis. This phenomenon is similar with the observation shown in Figure 4, where the stomata closure was greater induced in the sensitive M.toringoides, but not in the tolerant M.hupehensis. Both these two phenomenon suggest that the photosynthesis of M.toringoides and M.hupehensis were impaired, and the photosynthesis of the sensitive M.toringoides is more damaged by waterlogging stress.

Comment: Lines 223-224: Which two different roles does ethylene play in M. toringoides?

Response: Sorry for the misleading and we have seen that we are not sufficient for clearly explaining the roles of ethylene. According to our speculations based on the observation of different ethylene production under waterlogging stress in the two species, we thought that ethylene acts as a signal compound in M. hupehensis, while in M.toringoides ethylene acts as a signal compound at the initial stage of waterlogging stress and acts as toxic hormone when stress is prolonged. Now, we have rewritten the sentence, please see line 255-258.

Comment: Figure 6: The abbreviations in the legend must be defined in the figure caption since it is the first time some are used.

Response:According to your suggestion, we have added the explanation of the abbreviations in Figure 6 and checked all of the figures.

Comment: 2.6. The title “Transcriptome sequencing and read assembly” does not reflect the content of the paragraph. Any idea why more than 20% of the reads could not map to the reference?

Response: we have changed the title to ‘Transcriptome sequencing and mapping to the reference genome’. For the question about more than 20% of the reads could not map to the reference, the reason is that the apple reference genome (GDDH13 Version 1.1) is sequenced from ‘Golden Delicious’. Since the materials we used in the present work are apple rootstocks: M. hupehensis and M.toringoides, there must exist some differences in their sequences.

Comment: Figure 7A legend is incomplete.

Response: we have fully explained the Figure 7A in the revised manuscript, please see line 298-306.

Comment: In section 2.8 and 2.9 there is a lot of redundancy between the text and figures. It is better to highlight the most important observations while adding a sound perspective and leave the details to the figures or tables.

Response: according to your suggestion, we have rephrased these two sections, please see section 2.8 and 2.9.

Comment: 3.2. Title is not clear “Genes Involved in Flavonoids under waterlogging stress”

Response: we have changed it to ‘Differentially expressed genes involved in flavonoids biosynthesis under waterlogging stress’

Comment: 3.3. Title is not clear “Genes Involved in hormones under waterlogging stress”

Response: we have changed it to ‘Differentially expressed genes involved in hormone biosynthesis and signalling under waterlogging stress’

Reviewer 2 Report

Please recheck the sentences to improve English.

Please recheck the sentences to improve English.

Author Response

We are very appreciated for your positive response on our manuscript. According to your suggestion, our manuscript has been spell-checked and grammar-checked by Anita K Snyder, an English editor. We hope that the revised version of the manuscript is now acceptable for publication. Please see the attachment.

Thanks again.

Reviewer 3 Report

An interesting and well-conducted study on the differences in sensitivity to soil waterlogging stress in two apple rootstocks. The authors' investigations took into account the macroscopic aspects of the stress, some functional and metabolic components, and then a classical but careful transcriptomic study allowed the identification of differences in responses - in terms of transcripts - between two genotypes under two contrasting physiological conditions (maximum stress vs no stress). Overall, the results are convincing and clear, but the interpretation of a two-phase response to ethylene is delicate, and would have deserved to question the fluxes of ACC in the plant (root / shoot relationships); the transcriptomic analysis is rather convincing.
The discussion is a little too long but clear overall.

The final diagram, qualified as a model, is still insufficient to account for the processes at the temporal level and at the different scales (cell / organ / plant). A more conceptual diagram, showing the top- and down-regulations, would have been desirable.

A systematic checking of M. hupehensis spelling is required.

Main proposals of rewriting have been made in the joint file.

Author Response

Many thanks for your constructive comments, which were very valuable for improving our manuscript. We also appreciated for your editing on our English writing problem. We could see the problem of writing English, and according to your suggestion, we asked our colleague Anita K Snyder, an English editor, to have our revised manuscript spell-checked and grammar-checked.

We also have read the comments and suggestions carefully and revised the manuscript accordingly. We accepted and have made changes accordingly, which we marked in red in the revised manuscript. The yellow showed the changes of English writing. The minor part from your comments are directly corrected in the revised manuscript, while the part with your major concerns or questions were made in Point-by-point responses that are listed below. Hope that we answered your all concerns and we hope that the revised version of the manuscript is now acceptable for publication.

Point-by-point responses

Comment: 1. Apple (Malus domestica Borkh.) tree orchards are worldwide under threat of waterlogging stress, which affects the growth, quality and yield of apples and causes significant economic losses. This statement is too abrupt: in many cases the main stress is caused by drought. A brief review of the regional importance of waterlogging situations and their effects on crops would be desirable.

Response: we agree with reviewer’s comment and accordingly, we have rewritten this sentence to ‘Apple (Malus × domestica Borkh) is one of the most cultivated fruit crops in China. While apple tree frequently encounters waterlogging stress mainly due to excess rainfall, soil compaction, or poor soil drainage, resulting in yellowing leaf, declined fruit quality and yield in some region’.

Comment: The control and waterlogged seedlings of M.toringoides and M.hupenensis with uniform size were randomly selected and immediately transferred into a closed plastic box at room temperature 25℃. Are there some risks of root drying? Naked roots?

Response: It's not naked roots and no risks of root drying, since we put the pots of seedlings in the container, which does not cause any root damage or drying.

Comment: for the GC2010 Plus gas chromatography, precisions lacking: gas vector, T°, column, etc.

Response: The GC2010 Plus gas chromatography (Shimadz, Japan) equipped with an auto sampler and a flame ionization detector using a capillary GC column (Zebron ZB-WAX plus, 30 m, 0.25 mm ID, 0.25 µm). Helium was used as carrier and make-up gas. The injection port and detector temperatures were 240℃and 250℃, respectively. We have added this information into the respective part of manuscript, please see line 652-655.

Comment: 5. Is this sample of candidate genes sufficient?

Response: Since we were focused on the hormone related DEGS, we were then just selected the 6 six DEGs, which in our RNA-seq result were with highest expression. And the qRT-PCR results are consistent with our RNA-seq result. Therefore, we think this sample for qRT-PCR is sufficient for validation of RNA-seq. We also describe this aspect in the Results part, please see line 410-415.

Comment: This diagram is not clear enough, as it does not distinguish between the responses (positive vs. negative), nor the different scales (cellular metabolism vs. organ) likely to explain the tolerance or sensitivity responses of the young apple plants to soil waterlogging. Moreover, a temporal response to ethylene was suspected.

Response: Many thanks, now we have redrawn a model, where we compared the differences between M.toringoides and M.hupenensis. Please see Figure 12.